# Covering the Role of PGC-1α in the Nervous System

**DOI:** 10.3390/cells11010111

**Published:** 2021-12-30

**Authors:** Zuzanna Kuczynska, Erkan Metin, Michal Liput, Leonora Buzanska

**Affiliations:** Department of Stem Cell Bioengineering, Mossakowski Medical Research Institute, Polish Academy of Sciences, 02-106 Warsaw, Poland; zkuczynska@imdik.pan.pl (Z.K.); emetin@imdik.pan.pl (E.M.); mliput@imdik.pan.pl (M.L.)

**Keywords:** PGC-1α, central nervous system, mitochondrial biogenesis

## Abstract

The peroxisome proliferator-activated receptor-γ coactivator-1α (PGC-1α) is a well-known transcriptional coactivator involved in mitochondrial biogenesis. PGC-1α is implicated in the pathophysiology of many neurodegenerative disorders; therefore, a deep understanding of its functioning in the nervous system may lead to the development of new therapeutic strategies. The central nervous system (CNS)-specific isoforms of PGC-1α have been recently identified, and many functions of PGC-1α are assigned to the particular cell types of the central nervous system. In the mice CNS, deficiency of PGC-1α disturbed viability and functioning of interneurons and dopaminergic neurons, followed by alterations in inhibitory signaling and behavioral dysfunction. Furthermore, in the ALS rodent model, PGC-1α protects upper motoneurons from neurodegeneration. PGC-1α is engaged in the generation of neuromuscular junctions by lower motoneurons, protection of photoreceptors, and reduction in oxidative stress in sensory neurons. Furthermore, in the glial cells, PGC-1α is essential for the maturation and proliferation of astrocytes, myelination by oligodendrocytes, and mitophagy and autophagy of microglia. PGC-1α is also necessary for synaptogenesis in the developing brain and the generation and maintenance of synapses in postnatal life. This review provides an outlook of recent studies on the role of PGC-1α in various cells in the central nervous system.

## 1. Introduction

The proper function of mitochondria plays a pivotal role in the maintenance of neurons and neurodevelopment, including neuronal differentiation [1,2]. During stem cell differentiation, a metabolic switch occurs from glycolysis to oxidative phosphorylation (OXPHOS) for their energy source [3,4]. Single-cell transcriptomic studies confirmed that such a switch occurs during the neuronal commitment of stem cells, while neural stem cells (NSCs) reveal glycolytic metabolism, the metabolism of neurons dependent on OXPHOS [5]. Moreover, during the differentiation of NSCs, an increase in mitochondrial DNA (mtDNA) copy number and mitochondrial mass accompanying the metabolic switch is observed [6]. These increases are the results of mitochondrial biogenesis which is the process of production of new mitochondria from pre-existing mitochondria [7,8]. Therefore, these studies demonstrate the importance of mitochondrial biogenesis during the process of neurogenesis.

PGC-1α is a well-known coactivator of nuclear PPARs (peroxisome proliferator-activated receptors), ligand-activated transcription factors involved in mitochondrial biogenesis. Pioneering discovery of PGC-1α by Puigserver et al. investigating the process of adaptive thermogenesis revealed that its function is linked to mitochondrial biogenesis [9]. Further studies uncovered members of the PGC1 family: PGC-1α, PGC-1b, and PRC (PGC-1-related coactivator) [10]. PGC-1α is crucial for the proper metabolism in different types of tissue. It is extensively studied regarding its role in cardiac, muscle, and liver function and also in the fields of neurodegenerative diseases and mental disorders.

PGC-1α can be activated through many pathways that are not fully understood yet. Recently, Zehnder et al. revealed that upregulation of PGC-1α is controlled by metabotropic glutamate receptor 5 (mGluR5) in immature astrocytes [11]. However, there are two well-known pathways of PGC-1α activation: deacetylation by sirtuin 1 (SIRT1) or phosphorylation by AMP-activated kinase (AMPK). AMPK can activate PGC-1α directly through phosphorylation or indirectly through activation of SIRT1 by raising NAD+ levels [12]. Once the PGC-1α is activated, it translocates to the nucleus from the cytoplasm where it induces the expression of nuclear respiratory factor 1 (NRF-1) and NRF-2 and interacts with them via protein–protein interaction to increase their activities. As a result, NRF-1 and NRF-2 activate mitochondrial transcription factor A (TFAM) causing increased expression of nuclear genes encoding subunits of the five complexes in the mitochondrial electron transport chain (ETC) and further resulting in mtDNA replication and transcription. Therefore, the initiation of PGC-1α activation is the key element in the mitochondrial biogenesis process [13].

In the nervous system, high demands of energy for functioning make mitochondria essential, and PGC-1α is in the center of the interest as a regulator of mitochondrial biogenesis. Structural abnormalities in the different parts of the brain were observed in mice lacking PGC-1α [14,15]. Moreover, deficiency of PGC-1α in mice resulted in ataxia and alterations in gene expression in the cerebellum [16].

The dysfunction of mitochondria is present in many neurodegenerative diseases. Studies had shown that PGC-1α is involved in the pathophysiology of Huntington’s disease (HD) [17], Parkinson’s disease (PD) [18], and Alzheimer disease [19], amyotrophic lateral sclerosis (ALS) [20], and neuropsychiatric disorders such as schizophrenia [21,22] (reviewed in [23,24,25]).

Targeting PGC-1α may be beneficial for these diseases and disorders, but the comprehension of PGC-1α role and mechanism in the different types of cells in the nervous system must be achieved first to plan therapy strategy.

Therefore, in this review, we will focus on the role of PGC-1α in the functioning of different type of cells building the central and peripheral nervous system. We will also rise up the role of PGC-1α in synaptogenesis.

## 2. CNS-Specific Isoforms of PGC-1α

Apart from conventional PGC-1α transcripts in the brain, brain-specific isoforms were also discovered. They were detected in the total human brain RNA of four individuals by the combination of RNA ligase-mediated rapid amplification of cDNA ends (RLM-RACE) and long-range polymerase chain reaction (PCR) [26]. A new promoter of PGC-1α was identified and it is localized far upstream of the reference promoter of the PGC-1α gene [26] (Figure 1). This promoter initiates CNS-specific transcripts of PGC-1α, which differ from the reference gene transcript in N-terminus sequence and contain novel exons B1-B5 spliced to exon 2 of the reference gene [26,27]. These transcripts encode two full-length and several truncated isoforms of PGC-1α [26]. Full-length isoforms were detected in the nucleus, similarly to reference PGC-1α, whereas other forms were found in the cytoplasm or both localizations in SH-SY5Y cells using tagging with enhanced green fluorescent protein (eGFP) [26]. In mice, differences between the expression pattern of reference and CNS-specific isoforms of PGC-1α were detected only in the striatum [28]. Analysis performed on the post-mortem human brains showed differences in levels of expression of CNS-specific B1B4 and B5E2 transcripts depending on the region of the brain, while reference gene transcripts’ levels were less influenced by localization in the brain [29]. Specificity to the CNS of these transcripts suggests that these isoforms might take a part in the differentiation or preservation of neural cells. In the most recent studies, functional aspects of the CNS-specific transcripts were addressed. Activation of the CNS-specific PGC-1α promoter by Forkhead Box A2 (FOXA2), a transcription factor vital for differentiation and functioning of dopaminergic midbrain neurons but not the reference gene promoter in SH-SY5Y cells, indicates involvement of CNS-specific transcripts in the generation and maintenance of dopaminergic neurons [30]. It was shown that activation of CNS-specific PGC-1α promoter affected the expression of ALS and Parkinson’s disease risk genes in SH-SY5Y cells [27]. Still, the role of these isoforms in the functioning of the CNS or the pathophysiology of neurodegenerative diseases in vivo remains to be elucidated; therefore, it is too early to target them in therapies.

PGC-1α is differently regulated in different types of cells [31]. In the nervous system, the expression of PGC-1α depends on the region of the brain [28]. The comprehension of the functioning of PGC-1α and the effects of its disturbance in the different types of cells in the nervous system will provide better insight into the role of PGC-1α in neurodegenerative and neuropsychiatric diseases and will therefore enable the development of better treatment strategies.

## 3. Role of PGC-1α in Different Types of Neuronal Cells in the Nervous System

Particular types of neuronal cells are generally classified according to their morphology (multipolar, bipolar, and unipolar), physiology (sensory neurons, projection neurons, and interneurons), and function (inhibitory or excitatory). However, distinctions between these classes are not firm and their molecular or connectional properties frequently overlap (e.g., projection neurons include inhibitory spiny projection neurons but also excitatory glutamatergic motor neurons or cortico-thalamic neurons). On the top of this, complications of a unified nomenclature for neural cell types were lacking; thus, the modern form of hierarchical classification integrating morphological, physiological, and molecular (based on single cell transcriptomic data) categories was proposed by Zeng and Sanes [32]. In this review, by adopting the hierarchical classification approach, we will refer to individual regions of the nervous system, such as (1) brain (with special focus on cerebral cortex, midbrain, and cerebellum neurons), (2) retina, and (3) spinal cord, to further specify different neuronal cell types by their functional properties and their correspondence to PGC-1α expression or function.

### 3.1. Brain

Cerebral cortex is a highly organized brain structure containing cortical neurons. The main two classes of cortical neurons are glutamatergic excitatory neurons and GABAergic inhibitory neurons, with a variety of subclasses based on transcriptomic taxonomy [32]. Studies showed that PGC-1α expression is enriched in GABAergic interneurons expressing parvalbumin and glutamatergic projection neurons, especially in the cortical pyramidal layer V, which are the examples of inhibitory and excitatory neurons, respectively [21,24].

#### 3.1.1. Excitatory Neurons of Cerebral Cortex

It is well documented that PGC-1α is engaged in the regulation of transcription and excitability of neocortical and hippocampal excitatory neurons [21]. The data obtained in knockout experiments of PGC-1α in neocortical and hippocampal excitatory neurons revealed hyperactivity and enhanced glutamatergic transmission in neocortex and hippocampus, suggesting that PGC-1α is required for the regulation of both glutamatergic and fast-spiking interneuron populations [21]. Moreover, regional differences in dependence of PGC-1α were shown as hippocampal region-specific transcription levels were more reliable on the PGC-1α expression in comparison to neocortex [21]. Authors suggested that a more excitable state caused by the conditional knockout of PGC-1α might be a result of calcium buffer impairment and increased ROS production by dysfunctional mitochondria [21].

Motor neurons are projection neurons of the cerebral cortex, which can be classified under the subclass of pyramidal tract neurons as corticobulbar if terminated in brainstem or corticospinal if terminated in the spinal cord. The latter are responsible for a signal starting from the primary motor region of the cerebral cortex and carry information down to activate interneurons and lower motor neurons of spinal cord, which are innervating skeletal muscle cells. The primary motor region of the cerebral cortex contains cell bodies of the upper motor neurons, also referred to as giant Betz cells.

The generation and maintenance of motor neurons are dependent on the presence of PGC-1α from the early stages of their lifespan. During the differentiation of motor neurons, PGC-1α expression is increased, which is associated with the switch from glycolysis to OXPHOS metabolism [33]. In ALS, motor neurons are progressively lost. In post-mortem, human ALS spinal cord PGC-1α expression in motor neurons was decreased [20]. Similarly, in human ALS-derived motor neurons, in vitro levels of PGC-1α are downregulated [28]. The above results suggest that increased expression of PGC-1α might positively affect motor neurons survival. In ALS mice models, overexpression of PGC-1α protects from motor neurons loss [34,35]. Moreover, the lactate can improve the survival of motor neurons by the induction of CNS-specific PGC-1α [28]. Overexpression of PGC-1α in ALS iPSC-derived motor neurons resulted in the increased expression of mitochondrial complexes I and IV and prevented changes in axonal morphology [36].

The upregulation of PGC-1α levels in ALS motor neurons, which resulted in the reduction in their death, demonstrate the importance of PGC-1α in the normal functioning of motor neurons. In the future, PGC-1α might become a target for ALS therapies, although in the present, detailed molecular mechanism needs further investigation.

#### 3.1.2. Inhibitory Neurons of Cerebral Cortex and Cerebellum

Interneurons commonly function as connecting different types of CNS neurons or forming the key nodes within brain neural circuitry. PGC-1α, specifically, is localized in the nuclei of GABAergic interneurons, which form inhibitory synapses [37]. Evidence suggests that PGC-1α regulates gene expression in GABAergic interneurons and is essential for the normal functioning of these cells. PGC-1α deficiency reduced the expression of Ca^2+ –^binding protein parvalbumin in mice, which caused alterations in inhibitory signaling [38]. In the PGC-1α-deficient mice model, the balance between inhibition and excitation of synapses in the hippocampus was disrupted, caused by parvalbumin interneurons dysfunction [39]. Meanwhile, in another work, authors showed alterations of GABA inhibition in the motor cortex [40]. In the recent study, it was shown in mice lacking PGC-1α only in GABAergic that neurons parvalbumin expression was reduced in the cortex and hippocampus [41]. Behavioral tests conducted in mice demonstrated behavioral dysfunctions—a disorder of short-term habituation and overreacting to the stimulus [39]. The same group showed that mice lacking PGC-1α in the hippocampus also displayed a depression-like phenotype [42]. PGC-1α overexpression increases the expression of the GABARα2 subunit in the hippocampus and frontal cortex, causing anxiety-like changes in the behavior of the mice [43]. The disturbance of PGC-1α decreases parvalbumin-positive interneurons and results in the changed behavior of mice, but the detailed mechanism of these effects needs to be addressed in future studies.

Inhibitory signaling in CNS related to PGC-1α appears also in Purkinje cells, which are the type of projection neurons in cerebellum. PGC-1α knockout mice revealed deficiency of PGC-1α in the cerebellum, which negatively affected Purkinje cells, shown to be enriched with parvalbumin, similarly to inhibitory interneurons [16]. In 6-week mice, partial loss of Purkinje cells was noted and physiological properties of remaining cells were affected [16].

Together, these studies show the importance of PGC-1α in the proper functioning of inhibitory signaling in the brain.

#### 3.1.3. Midbrain Dopaminergic Neurons—Excitatory and Inhibitory Functions

Neurons producing neurotransmitter dopamine are primarily known to have excitatory properties by making glutamatergic synaptic connections but were shown to possess properties of dual excitatory and inhibitory functions. The latter is linked to the ability of controlling GABAergic transmission in a target-dependent manner and using plasma membrane uptake of GABA, without producing GABA internally [44].

Dopaminergic neurons are localized in the midbrain [45]. Knockout of PGC-1α in adult mice brain caused loss of dopaminergic neurons in substantia nigra [46]; thus, PGC-1α is necessary for the proper functioning of dopaminergic neurons and is engaged in the regulation of their viability. In another study, mice with dopaminergic neurons lacking PGC-1α were more predisposed to degradation of dopaminergic neurons in the presence of overexpressed α-synuclein, which was accompanied by ultrastructural changes of mitochondria and fragmentation of endoplasmic reticulum [47].

In the dopaminergic neurons, a few studies have shown growth factors interacting with PGC-1α, regulating its functions. Fibroblast growth factor-21 (FGF21) can activate PGC-1α through Nampt/SIRT1 pathway in the in vitro culture of human dopaminergic neurons [48]. Moreover, in midbrain neurons in vivo FGF21 improved mitochondrial functions by enhancing the expression of PGC-1α via AMPK activity [49]. In the context of PD, FGF21 could have a beneficial effect on survival of dopaminergic neurons i.a. through the activation of PGC-1α, which results in the improvement of mitochondrial functions [49]. The Glial Maturation Factor (GMF) was shown to be an upstream regulator of PGC-1α in rat immortalized dopaminergic neurons; however, it is not clear whether GMF represses PGC-1α expression directly or indirectly [50]. Studies performed by Khan et al. linked the deficiency of GMF in mice PD model with improved viability of dopaminergic neurons [51]. Connection between GMF-PGC-1α and GMF-PD suggests that further studies should be performed to investigate PGC-1α as a potential strategy against PD. Other non-dopaminergic neuronal systems are also involved in PD, such as serotonergic system. Dysfunctions of serotonergic system can cause motor symptoms, anxiety, and depression, diminishing the quality of a patient’s life [52]. Understanding interactions between serotonergic and dopaminergic system is important for the pathology of PD. As an example, dopamine replacement therapy with precursor L-DOPA is a widely used treatment for PD, resulting, however, in dyskinesias [53]. The reason behind it is that serotonergic neurons can convert L-DOPA to dopamine mediating dopamine release as a false transmitter. This shows that not only the dopaminergic system is involved in PD [52]. In the recent paper, serotonin was shown to activate Sirt1/PGC-1α via 5-HT_2A_ receptor. Serotonin-induced Sirt1, which is a key transcription factor in PGC-1α, had neuroprotective effect on cortical neurons challenged with excitotoxic and oxidative conditions [54].

Mitochondrial dysfunctions in the pathogenic process of PD were recently linked to the master regulator of mitochondrial biogenesis PGC-1α (reviewed in [55]). It was shown in several in vitro and in vivo models that overexpression of PGC-1α results in protection against neurodegeneration. Parkin (Parkinson juvenile disease protein 2), an enzyme protein encoded by PARK2 gene, of which mutations cause a recessive early-onset form of PD, controls PGC-1α expression and enhances mitochondrial biogenesis and survival of dopaminergic neurons [56]. The expression of PGC-1α in the human PD brain is also regulated by PARIS-zinc finger protein. It has been demonstrated that, in the *Drosophila* model, repression of PGC-1α mediated by PARIS leads to the loss of dopamine neurons, while overexpression of PGC-1α restores the functioning of mitochondria in the dopaminergic neurons by upregulation of NRF1 and TFAM [57]. Due to the role of PGC-1α in the maintaining of dopaminergic neurons, its protective effect was widely studied in PD. It was shown that drugs such as GW1929 [58], MitoQ [59], or roflupram [60], acting via the PGC-1α pathway, might be supportive for the preservation of dopaminergic neurons functioning in PD.

Even though PGC-1α is a very promising target in the therapies, induction of its overexpression might achieve an undesirable outcome. In the rodent model, overexpression of PGC-1α caused the degeneration of rat midbrain dopaminergic neurons [61]. In another study, overexpression of PGC-1α resulted in the vulnerability of mice neurons in substantia nigra to dopaminergic neurotoxin MPTP [62]. Moreover, overexpression of PGC-1α reduced the expression of Pitx3, which plays a key role in the development of dopaminergic neurons and led to the loss of the dopaminergic phenotype of neurons in substantia nigra [62]. On the other hand, more recent papers showed in the SH-SY5Y cell line that overexpression of PGC-1α improves protection against induced toxicity [63,64]. A positive effect of overexpressed PGC-1α was also observed in vivo in substantia nigra of C57BL/6 mice [45] and in the model of α-synuclein-induced degeneration in nigral dopaminergic neurons [47]. AAV-mediated overexpression of PGC-1α rescued αSyn-induced toxicity, restored basal respiration and mitochondrial morphology, and inhibited oxidative stress [47]. These studies strongly indicate that, when designing a therapy strategy, there is a need for careful adjustment of the level of PGC-1α expression to the physiological range.

#### 3.1.4. Role of PGC-1α in Synaptogenesis

Mitochondria in dendrites of neuronal cells are critical for the proper formation of the synapses and their functioning [65]. The formation of synapses was linked to the enhancement of mitochondria transport [66]. PGC-1α as the master regulator of mitochondrial biogenesis is also engaged in the process of synaptogenesis. It was shown that PGC-1α takes part in the development of dendritic spines and creating synaptic connections in the developing brain in mice. Increased PGC-1α levels enhanced the levels of synaptic proteins Synapsin 1 and PSD95 [67]. Moreover, PGC-1α is engaged in the developmental induction of synaptotagmin 2 and complexin 1 expression, which are calcium sensors crucial in the neurotransmitter release [68]. On another note, the negative impact of deficits of PGC-1α on synaptogenesis was linked with brain-derived neurotrophic factor (BDNF), which requires PGC-1α for synapses forming and upregulates PGC-1α involving ERKs and CREB [67].

Alterations of PGC-1α also influence the postnatal functioning of synapses. In the hippocampus of the adult brain, PGC-1α is necessary for the preservation of synapses [67]. Deletion of PGC-1α in astrocytes negatively influenced the postnatal generation of excitatory synapses due to the disruption of the mitochondrial network [11].

Taken together, PGC-1α is necessary for the proper generation and functioning of synapses in different types of CNS cells, but mechanisms still need elucidating, especially in the postnatal brain.

### 3.2. Dorsal Root Ganglia (DRG)

Sensory neurons are a part of peripheral nervous system (PNS) and they detect environmental stimuli and transfer them into the internal signal. The cell bodies of sensory neurons are localized in dorsal root ganglia (DRG), and project axons centrally into the ascending (sensory) dorsal white matter tracts of the spinal cord. Directly connecting lower motor neurons or through spinal cord inhibitory interneurons, they form sensory-motor neuron circuitry.

There are different types of specialized sensory neurons, e.g., mechanoreceptors, chemoreceptors, and hair cells.

Diabetic patients may develop diabetic peripheral neuropathy (DPN), which damages peripheral sensory neurons [69]. Downregulation of PGC-1α expression in dorsal root ganglia neurons of diabetic rats [70] suggests involvement of PGC-1α also in humans. In addition, it was shown in mice in vivo model that knock-out of PGC-1α causes mild neuropathy, which deteriorated in diabetic mice [71]. Moreover, overexpression of PGC-1α decreased oxidative stress through increased expression of antioxidant enzymes in DRG neurons in vitro [71]. These results indicate that PGC-1α can prevent neuropathy, mostly by prevention of oxidative stress. The protective effect of PGC-1α in sensory neurons was also shown in the zebrafish model. O’Donnell et al. demonstrated that expression of PGC-1α prevented α-Synuclein-induced toxicity and axonopathy in peripheral sensory neurons in vivo [72]. Additionally, the activation of the AMPK/PGC-1α signaling pathway by MT7 contributed to the regeneration of sensory neurons isolated from diabetic rat dorsal root ganglia [73]. Therefore, disturbance of PGC-1α expression deteriorates peripheral sensory neurons as a result of mitochondria damage, but overexpression or induction of PGC-1α may prevent apoptosis and increase regeneration. This implies the importance of PGC-1α in the regular functioning and maintenance of sensory neurons, although this subject needs further studies.

### 3.3. Spinal Cord

Interneurons of the spinal cord carry sensory information to the lower motor neurons. Most of the interneurons of spinal cord are found in grey matter (in Central Pattern Generator CPG circuits), including glutamatergic and inhibitory interneurons. The latter regulate motor activity by GABAergic signaling, but also releasing Glycine, important in spinal cord presynaptic inhibition. Spinal cord interneurons contribution into injury induced neuroplasticity is documented; however, this was not yet directly linked to PGC-1α activity and its role in this process still needs to be explained [74].

Elucidation of the PGC-1α’s role in the recovery after spinal cord injuries (SCI) was another studied direction. The contusive SCI generated by dropping weight at T10 spinal level in rodent model is followed at 7 days after injury by degeneration of mitochondria, decrease in PGC-1α expression, and activation of apoptosis in spinal cord neurons [75]. However, PGC-1α overexpression attenuates apoptosis of spinal cord neurons following SCI [75,76]. It was suggested that this effect might be associated with PGC-1α involvement in the downregulation of expression of molecules involved in RhoA-ROCK pathway—RhoA, ROCK1, and ROCK2 [75]. An increased level of PGC-1α was observed in the place of injury after 7 days of contusive SCI performed by dropping weight at T10 spinal level in rats, which indicates involvement of PGC-1α in the spontaneous regeneration of neurons [76]. This effect seems to be confirmed by the neuroprotective effect of tetramethylpyrazine, which may act through the activation of PGC-1α. Moreover, PGC-1α expression was localized mostly in the grey matter of the spinal cord, so the protective effect of PGC-1α might be mostly of neuronal origin [76]. In spinal cord neurons in vitro culture, small molecule ZL006 reduced ischemia caused apoptosis and protected mitochondria through the AMPK-PGC-1α-Sirt3 pathway [77]. Sirt3, which is a pivotal regulator of oxidative stress, is shown to be activated via PGC-1α [78]. Thus, the protective effect of PGC-1α in neurons might be mediated by the reduction in oxidative stress through the activation of Sirt3 expression.

Decreased level of PGC-1α after SCI in comparison to healthy tissue suggests PGC-1α involvement in the physiological functioning of the spinal cord [76]. The activation of PGC-1α in spinal cord neurons might be a promising target for the therapies of different spinal cord damages, but this possibility still needs further investigation to understand the molecular mechanism of PGC-1α functioning in spinal cord neurons.

Performing a movement requires a signal transmission from lower motor neurons to effector muscle fibers. Neuromuscular junctions (NMJ) are specialized synapses formed between motor neurons’ axons and muscle fibers, which use acetylocholine as a neurotransmitter in vertebrates [79]. The NMJ dysfunctions leads to the disturbance of motor function.

Overexpression of PGC-1α in all tissue of ALS mice models decreased degeneration of NMJ [34]. Actually, PGC-1α in myotubes activates a signaling pathway that leads to the generation of NMJ by motor neurons. The muscle PGC-1α expression leads to the pre-synaptic and post-synaptic remodeling of NMJ [80]. In the microfluidic system, muscle-specific PGC-1α1 induced signaling molecule neurturin, which led to the increased generation of NMJ between motor neurons and myotubes [81]. Moreover, targeting muscle PGC-1α may be beneficial for the mild impairment of NMJ [80]. In the future, ALS therapy strategies, not only neural-origin PGC-1α but also muscle-specific PGC-1α, might become targets to improve survival and functioning of motor neurons.

### 3.4. Retina

Retina is the part of the CNS originating as outgrowths of the developing brain. It is a multilayer, complicated neuronal tissue structure consisting of the several different cell types, connected to the brain by the optic nerve.

Photoreceptors are the type of sensory neurons localized in the retina, which convert light into an electrical signal. Studies performed by Egger et al. showed a high level of PGC-1α expression in the mouse retina [82]. The PGC-1α knockout resulted in the abnormal morphology and functioning of the retina [82]. Results obtained by this group indicated that PGC-1α plays a role in the repair processes in damaged retinal cells and the recovery of photoreceptors from light-induced damage [82]. Another group showed that repression of PGC-1α makes mice more vulnerable to external factors and aging, which resulted in the degeneration of photoreceptors [83]. Similarly, Rosales et al. also observed photoreceptors’ degeneration in mice with PGC-1α knockout [84]. Especially interesting is the fact that the role of PGC-1α in photoreceptors might go beyond protection and regeneration and take a part in functioning. In photoreceptor cells, PGC-1α expression is regulated daily by the light exposure, which suggests that PGC-1α mediates adjustment of phototransduction to the light changes through activation of relevant genes [85].

## 4. Role of PGC-1α in Glial Cells of Central Nervous System

Glia cells constitute the majority of all cells in the mammalian brain [86]. Glial cell types are of different developmental origin and include astrocytes, oligodendrocyte, and microglia lineage cells [87].

### 4.1. Astrocytes

Astrocytes are the most abundant glial cell type in the mammalian brain. They are involved in many brain functions such as a regulatory role in synapse formation (synaptogenesis) and further maintenance and efficacy of synapses [88]. Therefore, proper astrocyte maturation is essential for fine synapse formation.

Astrocyte maturation occurs during the first 3 postnatal weeks in the developing rodent brain. After the first week of postnatal weeks, the astrocytes’ replication comes to an end and astrocyte maturation begins [89]. During the postnatal development, short-term upregulation of pathways associated with regulation of mitochondrial biogenesis and function has been divulged by transcriptome analysis of astrocytes [90]. Proper mitochondria activity is important for healthy astrocyte maturation. One particular regulator of mitochondria activity and mass is PGC-1α [91].

The recent study of Zehnder et al. indicated an important role of mitochondrial biogenesis in the regulation of astrocyte maturation and synapse formation [11]. The genetic silencing of mitochondrial biogenesis by deletion of PGC-1α resulted in the extended proliferation and hindrance of astrocyte morphogenesis. Their results showed that PGC-1α activity is needed for the transition from proliferation state to maturation state during astrocyte development. A defective mitochondrial network causes impaired maturation consequently influencing cortical development [11]. In the earlier study of Augustyniak et al., in human iPSC differentiating into neurons, activation of mitochondrial biogenesis was linked to upregulation of the expression PPARGC1A gene, encoding PGC-1α and neuronal to glial fate decision switch [8]. Overall, this data indicate that PGC-1α is indispensable for the generation and maturation of astrocytes due to its role as mitochondrial biogenesis activator.

### 4.2. Oligodendrocytes

Oligodendrocytes are neuroglia, whose role is to support neurons by myelination in the CNS. Myelination of axons allows rapid neural signal conduction and maintenance of axonal integrity [92]. Oligodendrocytes form myelin sheaths wrapped around axons to ensure normal neural signal conduction. They also metabolically support axons.

Involvement of PGC-1α in myelination was first shown by Cowell et al. [37]. They showed that during myelination there is a considerable increase in PGC-1α level in rat brain. Later on, Xiang et al. showed that PGC-1α is involved in postnatal axonal myelination and that deficient PGC-1α activity in oligodendrocytes may contribute to abnormal myelination in HD [93]. They found a decrease in myelin basic protein (MBP) in the striatum of PGC-1α KO mice which was accompanied by a decrease in brain cholesterol (MBP precursor) and the rate-limiting enzymes for cholesterol synthesis (HMG CoA synthase and HMG CoA reductase). It was shown that PGC-1α binds to the MBP promoter and its overexpression led to increased MBP and SREBP-2 promoter activity. These results support the involvement of PGC-1α in myelination through MBP expression as well as in cholesterol metabolism in oligodendrocytes. Moreover, overexpression of mutant huntingtin (Htt) in primary oligodendrocyte culture resulted in a reduced level of PGC-1α and its targets, which was also observed in mouse HD model expressing full-length mutant Htt [93]. In several HD models, reduced cholesterol biosynthesis and dysfunction of PGC-1α were reported [94,95]. Though reported PGC-1α dysfunction in HD was shown to have effects on neurons and muscle cells, Xiang et al. showed that inhibition of PGC-1α by mutant Htt in oligodendrocytes is another course of action in the pathogenesis of HD [93].

In another study, Xiang and Krainc demonstrated a therapeutic target for correction of deficient myelination in HD [96]. They showed that overexpression of SIRT1 or treatment with resveratrol (RSV) increased PGC-1α expression and rescued myelination and oligodendrocyte differentiation in oligodendrocytes expressing mutant Htt [96]. In several neurodegenerative diseases, including multiple sclerosis (MS), oligodendrocytes are targeted by demyelination which impairs the transmission of neural impulses and global synchronization of neuronal circuit activity. It can be prevented by multistep remyelination processes, which are initiated after demyelination started. Previous studies showed that regular physical activity stimulates anti-inflammatory immunomodulation, reduces brain lesion volume, and delays disease onset in animal models of MS and patients with MS [97,98]. Jensen et al. demonstrated that immediate exercise following lysolecithin-induced demyelination in mice enhances oligodendrogenesis, the rate of remyelination, and the proportion of remyelinated axons [99]. Transcriptomics analysis suggested broad activation of pro-remyelination pathways and activation of PGC-1α, a key regulator of energy metabolism [99]. Immunohistochemistry analysis confirmed that, during the differentiation of oligodendrocytes in early remyelination, PGC-1α is translocated into the nucleus where it co-localizes into nuclear complexes, followed by rapid degradation [99]. Exercise-induced activation of oligodendroglial PGC-1α led to enhancement of myelin thickness [99]. This was accompanied by clearance of lipid debris from lesions. Exercise combined with a remyelinating medication clemastine further enhanced remyelination of lesions in mice [99]. The study conducted by Yoon et al. supports the link between exercise and PGC-1α activation promoting myelinogenesis. Moreover, the authors showed inclusion of high dietary fat combined with exercise training has the great capacity in PGC-1α induction, which is correlated with MBP production resulting in promotion of myelinogenesis in the adult spinal cord [100]. These studies may help to understand the role of PGC-1α with the effects of high dietary fat intake and exercise in enhanced remyelination and protection to oligodendrocyte progenitor cells (OPC) and oligodendrocytes.

PGC-1α was shown to be involved in the protection of OPC in inflammatory conditions in vitro. De Nuccio et al. showed that pioglitazone-proliferator-activated receptor-γ (PPAR-γ) agonist protects OPC against tumor necrosis factor-alpha-induced damage by induction of key genes involved in mitochondrial biogenesis: PGC-1α, UCP2, and COX1 [101]. Authors suggest that this mechanism could be an effect either of positive autoregulatory loop of control of the PGC-1α gene through coactivation of PPAR-γ or an alternative scenario where pioglitazone-induced mitochondrial biogenesis was facilitated by other transcription factors.

### 4.3. Microglia

Microglia are the major cell type of CNS with immune function. They are crucial in pathological events such as immune surveillance and response to brain injury. After stroke, microglia cells act as early immune responders, which leads to the initiation of inflammation processes [102,103]. Han et al. showed that increased expression of PGC-1α was an immediate response to the ischemic stroke in the acute ischemic stroke mouse model and human patients [104]. Overexpression of PGC-1α in microglia led to suppression of neurologic deficits after ischemic injury, which was accompanied by the induction of mitophagy and autophagy. PGC-1α-induced autophagy and mitophagy resulted in the inhibition of inflammatory response through suppression of NLRP3, which has a vital role in many inflammatory diseases, such as autoimmune diseases [104,105]. Microglia can alter their phenotypes and functions in response to changes in a micro-environment. Depending on the predominance of secreted factors, microglia have functional plasticity and dual phenotypes-classic activated phenotype M1 (proinflammatory) and alternative activation phenotype M2 (anti-inflammatory). Yang et al. demonstrated that resveratrol promoted microglia polarization toward the M2 phenotype, which resulted in suppression of microglia activation and reduced LPS-induced inflammatory damage [106]. Knockdown of PGC-1α hampered M2 polarization and inhibited NF-κB and STAT-3/6 signaling, which suggests the role of PGC-1α is in resveratrol-induced microglia M2 polarization [106]. Previous studies showed that activation of microglia-specific cannabinoid type 2 receptor (CB2R) suppressed neuroinflammation in microglial cells. Ma et al. demonstrated that AM1241-induced activation of CB2R promoted M2 polarization, induced expression of BDNF, mitochondrial biogenesis markers, and reduced expression of pro-inflammatory markers [107]. Knockdown of PGC-1α reversed AM1241-induced anti-inflammatory process which demonstrated that PGC-1α-induced mitochondrial biogenesis is involved in cannabinoid CB2R agonist-induced M2 polarization [107]. These studies showed role of PGC-1a in suppression of immune response and microglia activation after brain injury.

## 5. Conclusions and Future Perspectives

PGC-1α is crucial for proper functioning of the variety of cell types across the central nervous system. PGC-1α is connected with the inhibitory signaling via parvalbumin as it was shown by the animal studies correlating lower expression of PGC-1α with the expression level of parvalbumin. These alterations transmit to the balance between inhibition and excitation and circuit function. Thus, PGC-1α is predominantly needed for the proper inhibitory neurotransmission, showing the importance of PGC-1α in healthy functioning of the nervous system. Moreover, impairment of functioning or decreased viability of neuronal cells may be linked with the dysfunction of mitochondria as an effect of disturbed PGC-1α expression.

The cell-specific targeting of PGC-1α in glial cells could be a promising and attractive therapeutic approach. This strategy could allow restoration of damaged cortical networks, induction of synaptogenesis, and remyelination in many debilitating neurodevelopmental and neurodegenerative diseases associated with dysfunction of glial cells.

Moreover, PGC-1α might have a positive effect on the functioning and viability of the central nervous system cells by activating mitochondrial biogenesis and reducing oxidative stress through activation of oxidative stress regulators such as Sirt3. It is also implicated in activation of function-related genes responsible for neuroprotection. The hypothesis of the crucial role of PGC-1α in neuroprotection already has strong experimental support but needs further investigation from human in vitro disease models (e.g., human iPSC-derived brain organoids) and clinical data. Moreover, PGC-1α is mostly studied through the prism of neurodegenerative diseases, so its detailed role in a healthy nervous system needs addressing. In Figure 2, the key roles of PGC-1α in neuronal and glial cells across the central nervous system are summarized.

Many studies showed that PGC-1α is a promising target for neurodegenerative diseases. It can be targeted through activation and overexpression or, in some cases, downregulation. The PGC-1α’s positive influence on the mitochondria in the nervous system and its neuroprotective effect was experimentally proved in vitro and in vivo. In the future even if PGC-1α might not be the main target in therapy, as a part of supporting therapy, it will lead to the improvement of life for many patients.

## Figures and Tables

**Figure 1 cells-11-00111-f001:**
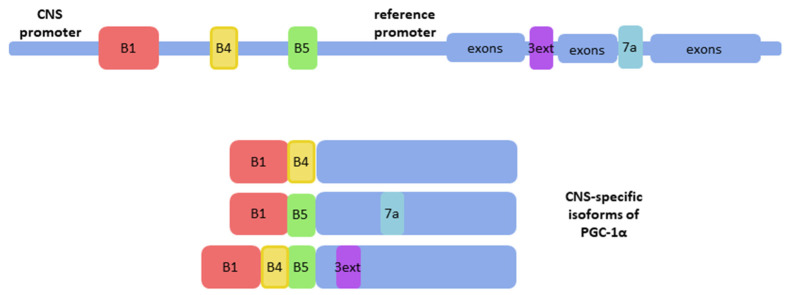
CNS-specific isoforms of PGC-1α include exons localized upstream to the reference promoter; B1, B4, B5: CNS-specific exons (red, yellow, and green respectively); 7a and 3ext: alternatively spliced transcripts encoding stop codons in exon 7a or in an extension of exon 3 (blue and purple, respectively) (based on [26] and [27]).

**Figure 2 cells-11-00111-f002:**
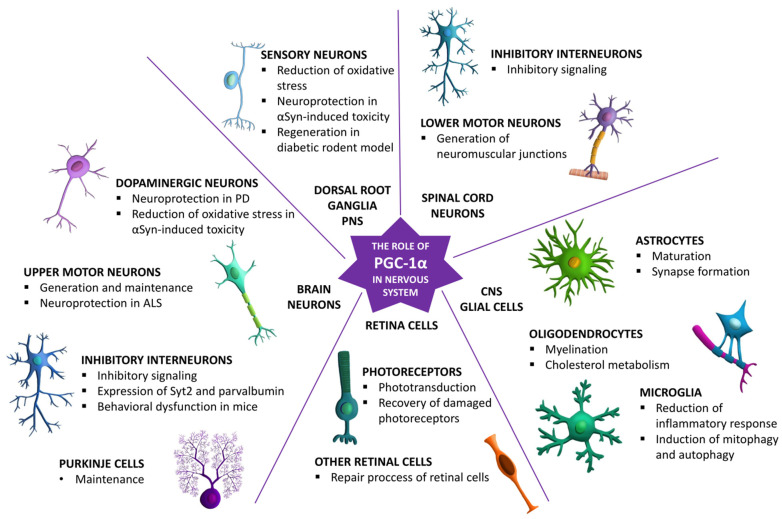
The key roles of PGC-1α in the maturation, functioning, and survival of different cell types across the nervous system.

## Data Availability

Not applicable.

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
