# Peer review of "Covering the Role of PGC-1α in the Nervous System"

_cells, 2021, doi:10.3390/cells11010111_

Round 1

Reviewer 1 Report

The manuscript by Zuzanna Kuczynska, Erkan Metin, Michal Liput, and Leonora Buzanska, entitled “Covering the role of PGC-1α in the Nervous System”, submitted to CELLS, concerns an important subject of the expression and regulation of the peroxisome proliferator-activated receptor-γ coactivator-1α (PGC-1α) in the nervous system. This molecule is a well-known transcriptional coactivator involved in mitochondrial biogenesis. Its activity puts that molecule in the center of the cellular metabolism. As the Authors state in the Abstract of their manuscript, PGC-1α is implicated in the pathophysiology of many neurodegenerative disorders, therefore a deep understanding of its functioning in the nervous system may lead to the development of new therapeutic strategies.

The  manuscript is divided into several chapters, with well written Introduction describing PGC-1α function linked to mitochondrial biogenesis, signaling pathways involved in the regulation of the molecule, and dysfunctions of the nervous system related to abnormal PGC-1α expression level.  

Chapter 2, which describes CNS-specific isoforms of PGC-1α would benefit from more precise description of the animal species and material derived from human subjects, which were the source of  PGC-1α (e.g. line 81-82, 92), and methods of analysis, with a comment on the comparative analysis of the material from different sources.  Abbreviations should be explained on the first appearance basis (OXPHOD, FOXA-2).

The subsequent CHAPTERS raise the majority of my questions and criticisms.

  1. Firstly, why the authors start their description of the CNS from the INTERNEURONS and not from the PROJECTION NEURONS, in which PGC-1α is linked to ALS, and to peripheral neuropathies, extensively discussed in the following chapters?

I am sure that the Authors had their justification of such an arrangement; it would be useful to hear on that. Anomalies which were historically described first? The most pronounced phenotypic changes? The abundance and completeness of the data available? Perhaps a figure showing interdependence of the brain data would be of some value?

MCMeekin and coworkers, who published this year an article (in Cells) on Dysregulation of PGC-1α-Dependent Transcriptional Programs in Neurological and Developmental Disorders: Therapeutic Challenges and Opportunities describe, that using an antibody for PGC-1α and, subsequently, small molecule fluorescent in situ hybridization they observed the enrichment of PGC-1α expression in neurons which express glutamic acid decarboxylase 67 and the calcium buffer parvalbumin (PV) -  so perhaps that would be a starting point? That is only the suggestion because having a knowledge on the level of gene and protein expression in particular cell populations the Authors might consider their values as predictors of the PGC importance. And then look at different fast-firing neuron populations.

Coming back to the content:  The first, introductory paragraph of 3.1 chapter is redundant and not very precise (additionally to GABA INT release Glycine, important in spinal cord presynaptic inhibition). Purkinje cells, which are mentioned in that chapter, are not interneurons: these cells project to the deeper cerebellar nuclei.

  1. Secondly, there is a description of the dopaminergic neurons followed by UPPER MOTONEURONS. I would strongly suggest to reconsider this, and describe the cortical neurons and subcortical neurons instead. This way there is a space to include e.g. very interesting regulation of  PGC-1α in cortical neurons (not specifyung whether they derive from pre-motor or motor areas) by serotoninergic fibers, as reported recently (see Sashaina E. Fanibunda et al. Serotonin regulates mitochondrial biogenesis and function in rodent cortical neurons via the 5-HT2A receptor and SIRT1–PGC-1α axis, PNAS 2019). Such approach allows to look at the brain as the network. 
  2. Regarding LOWER Motoneurons: they do not belong to the PNS. I comment on this and other classifications with reference to Figure 2
  3. In CHAPTER 5.2. „OLIGODENDROCYTES” the authors concentrate their attention on the involvement of PGC-1α in postnatal axonal myelination. There are several useful data gathered here, albeit some information is worth a comment – like that PGC-1α KO mice reveal a decrease in MBP in striatum. What do we know and what is our understanding of that? Are striatal Oligo mechanisms and MBP vulnerable in particular to that deficit?

 It is my general comment that the review lacks some unifying comments or discussion (could be extracted from  the authors of the publications quoted here) which would put the observations in a functional context and make that article more juicy.

There is an interesting set of data published (De Nuccio et al., Exp. Neurol. 2015), which shows that Peroxisome proliferator activated receptor-γ agonists protect oligodendrocyte progenitors against tumor necrosis factor-alpha-induced damage.  Kuczynska et al. do not quote that article, which, in my opinion, is worth to be included. The reported protection was found to be related to the effects on mitochondrial functions and differentiation and associated with PGC-1α increased expression.

Kuczynska et al., begin their review article with the statement on that PGC-1α is a transcriptional coactivator involved in mitochondrial biogenesis; perhaps adding this article to the reviewed literature would strengthen the link of PGC-1alpha and mitochondria-mediated cellular effects. I want to draw the attention of the Authors to discrimination between the effects related to OPCs and those related to mature oligodendrocytes.

  1. Also, I did not find quotation of the paper by Yoon et al., (Scarisbrick’s lab, Mayo Clinic, BBA 2016) which presents the data on the interplay between exercise, dietary fat and myelinogenesis in CNS – associated with PGC1-alpha activation (See Fig. 7. High dietary fat and exercise converge on SIRT1 and PGC-1αto modulate mitochondrial abundance in the adult spinal cord). I am convinced that would enrich the thread on the role of physical activity in myelination and PGC-1 alpha.

6. Figure 2. The concept to summarize the role (proved or suggested) of PGC-1 alfa with respect to different types of neurons and glial cells building the central/peripheral nervous system is good. However, (1) there are several serious mis-statements (repeating those in the text), which mislead the reader, (2) the classification of the categories of PGC-1 alpha biological activity is unclear.

In my opinion showing the convergence of intracellular pathways in different types of cells leading to protection and metabolism activation (like myelinization in OLIGOS and synaptic plasticity in NEURONS) would provide more comprehension. Also using colour scale to mark levels of expression (if possible to extract from the literature) would be the added value.

However, if this Figure is to be included in the article, it requires several corrections and rearrangement. Namely:

  1. Classifying cell types: 1A. Despite its peripheral location, the retina (and its photoreceptors)  is a neural portion of the eye, thus it is actually part of the central, not peripheral nervous system.   During development, the retina forms as an outpocketing of the diencephalon.  1B.There are two main types of MNs, (i) upper MNs that originate from the cerebral cortex and (ii) lower MNs that are located in the brainstem and spinal cord. Both belong to CNS.

Thus neither retinal cells not lower motoneurons belong to the peripheral nervous system, as drawn on the graph.

2.Distinguishing the category of activity:

E.G. SENSORY CELLS BOX.

“protection of  retinal cells” and “neuroprotection”: what is the difference in the range of action between these two??? Retinal cells belong to neuronal cells. Perhaps it would be better to use the general term NEUROPROTECTION – and then list types of cells which were shown to be protected by PGC-1 alfa. If there is a cell specificity, please show it.

Moreover, in the same category of SENSORY NEURONS the authors list “functioning of photoreceptors”. What does it mean? Again – divide these information into: taste receptors, photoreceptors etc.

3. UPPER MOTONEURONS – generation of neuromuscular junctions. This information if far from the knowledge we have. There are many classic papers showing the connections of corticospinal tract (both from neurons in the premotor and motor cortex) to the spinal cord areas, but there is no data on the connections of the upper motoneurons to the muscle (targets which would influence NMJ)?. Recent review By Olivares-Moreno et al, published earlier this year (Front. Neurosci., 11 June) is a valuable review of the current status on our knowledge on CS axons terminal targets:

 CS terminations densely innervate the dorsal horn and intermediate zone of the gray matter in the spinal cord, which is probably related to the descending control of sensory afferent inputs and limb muscle control (Lemon 2008). Despite a certain highly conserved organization of the CS tract through different species, there are important variations related to fine movement of the extremities and digits, since these types of movements emerged at different times throughout evolution (Iwaniuk and Whishaw, 2000; Lemon, 2008). One of the most notable differences is the presence or absence of monosynaptic cortico-motoneuronal (CM) projections between primate and non-primate species (Rathelot and Strick, 2006). (….) some studies show that there are no CM connections in species such as mice, rats, raccoons, and cats, even in primates such as marmosets or lemurs (Illert et al., 1976; Gugino et al., 1990; Yang and Lemon, 2003; Alstermark and Ogawa, 2004; Lemon and Griffiths, 2005). So there is a strong experimental proof that in many species the monosynaptic pathway from the upper motoneuron to the lower one is scarce or does not exist. So what is a mechanism that would lead to altered generation of NMJ which are terminals of the lower MNs? I was trying to refer to the text of the work, but I did not find a mention on that. I assume it requires explanation.

  1. In general, I would propose to add to the Legend of this summarizing figure a statement: the data were extracted from… (or the graph is based on the data from…) and specific numbers of the references will follow)

7.In CHAPTER 6. ROLE OF PGC-1alpha IN RESPONSE TO SPINAL CORD INJURY more precision in description of the data is advised. The postlesion changes, their scope and severity depend on the time and distance from the injury site. Therefore all the data need to be described with precision, which was a type of injury (complete or incomplete transection, contusion, hemisection etc), perilesion area or remote area etc.

Reviewer 2 Report

Dear authors,

The review is interesting for the biomedical community, especially for those teams focused on mitochondrial function in the CNS. I have minor comments: for part 2, it will be excellent to have a graph with the specific isoforms/haplotypes in each of the brain structures based on the reviewed literature.  In lines 317 and 318 please correct the expression of mutant huntingtin (Htt). In line 322, mutant huntingtin should be mutant Htt. Please recheck the information between lines 346-348 as authors in reference 89 mentioned “Microglia-specific PGC-1α overexpressing mice exhibited significantly decreased neurologic deficits after ischemic injury, with reduced NLRP3 activation and proinflammatory cytokine production.” In several lines of the text 264, 377, and 405, the authors mention that PGC-1α protects from oxidative stress, are there any insights into why producing more mitochondria helps to mitigate the damage induced by ROS?

The conclusion paragraph and phrases are repetitive in each section. It could be highly appreciated, to deepen the analysis of the conclusions of each reviewed paper into “mechanistic” and molecular insights of the regulation of PGC-1α. In essence, not be too general. 

Eager to receive the corrected version!
